# Elevated CTRP1 Plasma Concentration Is Associated with Sepsis and Pre-Existing Type 2 Diabetes Mellitus in Critically Ill Patients

**DOI:** 10.3390/jcm8050661

**Published:** 2019-05-11

**Authors:** Eray Yagmur, David Buergerhausen, Ger H. Koek, Ralf Weiskirchen, Christian Trautwein, Alexander Koch, Frank Tacke

**Affiliations:** 1Medical Care Center, Dr. Stein and Colleagues, D-41169 Mönchengladbach, Germany; 2Department of Medicine III, RWTH-University Hospital Aachen, D-52074 Aachen, Germany; david.buergerhausen@rwth-aachen.de (D.B.); ctrautwein@ukaachen.de (C.T.); akoch@ukaachen.de (A.K.); ftacke@ukaachen.de (F.T.); 3Section of Gastroenterology and Hepatology, Department of Internal Medicine, Maastricht University Medical Medical Centre (MUMC), 6202AZ Maastricht, The Netherlands; gh.koek@mumc.nl; 4Institute of Molecular Pathobiochemistry, Experimental Gene Therapy and Clinical Chemistry, RWTH-University Hospital Aachen, D-52074 Aachen, Germany; rweiskirchen@ukaachen.de; 5Department of Hepatology and Gastroenterology, Charité University Medical Center, D-10117 Berlin, Germany

**Keywords:** C1q/TNF-related protein 1, CTRP1, ICU, sepsis, inflammation, diabetes, critical illness, glucose metabolism, adipokine, metabolism

## Abstract

The adipokine family of C1q/TNF-like proteins (CTRP) plays a critical role in regulating systemic energy homeostasis and insulin sensitivity. It is involved in pathophysiological processes including inflammation and insulin-resistant obesity. Sepsis is associated with metabolic alterations and dysregulated adipokines, but the role of CTRP1 in critical illness and sepsis is unclear. We investigated CTRP1 plasma concentrations in 145 septic and 73 non-septic critically ill patients at admission to the medical intensive care unit (ICU) in comparison to 66 healthy controls. We also assessed associations of CTRP1 with clinical characteristics, adipokine levels, metabolic and inflammatory parameters. CTRP1 plasma concentration was significantly elevated in critically ill patients compared to healthy subjects. CTRP1 levels were significantly higher in ICU patients with sepsis. CTRP1 correlated strongly with markers of inflammatory response, renal function, liver damage and cholestasis. Furthermore, CTRP1 levels were higher in ICU patients with type 2 diabetes mellitus, and correlated with HbA1c and body mass index. This study demonstrates significantly elevated levels of CTRP1 in critically ill patients, particularly with sepsis, and links circulating CTRP1 to inflammatory and metabolic disturbances.

## 1. Introduction

The highly conserved family of secreted C1q/TNF-related (glyco-)proteins (CTRP) is a paralogue of adiponectin with diverse functions in regulating metabolism and immunity [1,2,3,4]. Next to adiponectin, 15 additional CTRP family members have been identified to date [3]. The CTRP family is involved in a variety of clinical and pathophysiological processes including immune defense, inflammation, apoptosis, autoimmunity, cell differentiation, organogenesis and insulin-resistant obesity [5].

CTRP1 is a novel member of the CTRP family secreted by different tissues, mainly adipose tissue [2,6,7]. It is known that CTRP1 plays a critical role in regulating systemic energy homeostasis and insulin sensitivity [2,4,7,8,9]. For maintaining proper balance of energy substrate metabolism, CTRP1 depends on complex metabolic networks of secreted hormones (e.g., insulin, leptin, adiponectin) [4]. CTRP1 is reported to be involved in PI3K (phosphatidylinositol 3-kinase)/Akt (protein kinase B) signaling pathways to induce glucose transport by insulin [9,10]. Increase in Akt signaling affects glucose metabolism by increasing translocation of glucose transporters GLUT1 and GLUT4 to the plasma membrane and by activating glycolysis enzymes indirectly [11,12].

It is well known that infectious and inflammatory diseases such as sepsis and systemic inflammatory response syndrome are accompanied by metabolic alterations such as insulin resistance or dysregulated adipokines [13]. Conversely, metabolic diseases such as visceral obesity and type 2 diabetes are characterized by high levels of pro-inflammatory cytokines [3,14]. Regarding adipokines in critical illness and sepsis, these factors have gained great attention as potential biomarkers for disease severity and as potential targets for therapy. While several “inflammatory adipokines” are upregulated in critical illness, including visfatin or resistin [15,16], a surprisingly high number of circulating adipokines, such as leptin, omentin or adiponectin, do not differ between healthy controls and critically ill patients [13,17,18]. Nonetheless, even if not different from healthy controls, circulating adipokines like omentin or adiponectin have been associated with survival in critical illness [17,18].

Furthermore, CTRP1 is reported to be involved in blood pressure regulation and cardiovascular function [9,10]. In addition to effects in glucose metabolism, the CTRP1-dependent Akt signaling activates the Ras homolog gene family (Rho)/Rho kinase (ROCK) signaling pathways resulting in aldosterone secretion and vasoconstriction [9]. Moreover, the stimulation of human vascular smooth muscle cells by CTRP1 results in upregulated expression of pro-inflammatory cytokines such as interleukin 6 (IL-6), monocyte chemoattractant protein 1 (MCP1) and intracellular adhesion molecule 1 (ICAM1) [19]. These genes were proposed as potential key modulators regarding the function of CTRP1 in inflammatory diseases [19].

Critically ill patients are closely associated with the entire spectrum of metabolic-related disorders, including diabetes and inflammation such as sepsis and severe inflammatory response [6]. In sepsis, the role of CTRP1, with regard to its critical role in regulating systemic energy homeostasis and insulin sensitivity, is currently not fully understood. Moreover, the potential value of circulating CTRP1 in critically ill patients is unknown. We therefore assessed plasma CTRP1 concentration in a large cohort of critically ill medical patients admitted to the intensive care unit (ICU) in order to investigate the diagnostic and clinical relevance of circulating CTRP1.

## 2. Experimental Section

Critically ill patients were included at admission to the medical ICU at the RWTH University Hospital in Aachen, Germany. The current cohort of patients was collected from an ongoing, prospective observational trial in our ICU at the RWTH University Hospital, in which patients were included consecutively. For the current analysis, we randomly enrolled *n* = 218 patients that had been treated between 2006 and 2011 from the existing biobank. Patients who were admitted for post-interventional observational stay or underwent an elective procedure were excluded, according to an established protocol [20]. The patients were categorized as sepsis and non-sepsis according to the Third International Consensus Definitions for Sepsis and Septic Shock (Sepsis-3) [21], and were treated following the current guidelines for treatment of sepsis (Surviving Sepsis Campaign) [22]. As a healthy control group, we analysed *n* = 66 blood donors (43 male, 23 female, median age 29.5 years, range 18–67 years, BMI median 25.4 kg/m^2^, range 17.9–37 kg/m^2^) with normal blood counts, normal values of liver enzymes and a negative serology for viral hepatitis and HIV [23].

The local ethics committee approved the study in accordance to the ethical standards laid down in the Declaration of Helsinki (reference number EK 150/06). All included participants provided written informed consent.

Blood samples were collected at the time of admission (before specific therapeutic measures), centrifuged and plasma was stored at −80 °C. Plasma CTRP1 concentrations were determined using a quantitative sandwich enzyme immunoassay (ELISA), according to the manufacturer’s instructions (Human CTRP1, #RD191153100R, BioVendor, Brno, Czech Republic). A repeat measurement for CTRP1 concentrations outside the linearity was not performed due to the small sample volumes available. Pre-dilution was used instead (20-fold). Thus, only a few patients had CTRP1 concentrations (*n* = 23) above linearity. CTRP1 concentrations above the linearity of the standard curve (1600 ng/mL) were set to 1600 ng/mL (corresponding to the highest CTRP1 concentration in the standard preparations) in order to minimize accidental overinterpretation of the data. All samples were included in the statistical analyses.

Owing to the skewed distribution of the parameters, data are given as median and range, and shown graphically by box-and-whiskers plots. They show a summary of the median, quartiles, range and extreme values. Their whiskers range from the minimum to the maximum value, excluding outliers displayed as separate points. An outlier was defined as a value that is smaller than the lower quartile minus 1.5-times interquartile range, or larger than the upper quartile plus 1.5-times the interquartile range. A far out value was defined as a value that is smaller than the lower quartile minus three times the interquartile range, or larger than the upper quartile plus three times the interquartile range. The degree of association between two variables was assessed by the Spearman rank correlation test. Comparisons of parameters between two different groups were conducted with the Mann–Whitney U-test. All values, including outside values as well as far out values, were included. *p*-values less than 0.05 were considered as statistically significant. All analyses were performed with IBM SPSS Statistics (SPSS; Chicago, IL, USA).

## 3. Results

### 3.1. CTRP1 Plasma Levels Are Significantly Elevated in Critically Ill Patients as Compared with Healthy Controls

Our study cohort comprised of 218 patients that had been admitted to the medical ICU. About two thirds of the patients were mechanically ventilated, the median Acute Physiology and Chronic Health Evaluation (APACHE II) score was 18 and 22.5% died during the course of ICU treatment (Table 1), thereby representing a cohort of critically ill patients with a high risk profile. CTRP1 plasma levels were significantly elevated in this cohort of critically ill patients (median 747.1 ng/mL, range 200.5–1600.0 ng/mL; Table 1) at admission to the ICU, as compared with 66 healthy controls (median 316.3 ng/mL, range 171.1–1308.7 ng/mL, *p* < 0.001; Figure 1a).

### 3.2. Elevated CTRP1 Plasma Levels in Critically Ill Patients Are Associated with the Presence of Sepsis

Within the cohort of ICU patients, plasma concentrations of CTRP1 were significantly increased in patients with sepsis (*n* = 145, median 779.6 ng/mL, range 200.5–1600.0 ng/mL) as compared to patients without sepsis (*n* = 73, median 574.2 ng/mL, range 227.2–1600.0 ng/mL, *p* = 0.006; Figure 1b). Typical sites of infection in sepsis were pneumonia, abdominal and urogenital tract, while non-sepsis causes of critical illness included, among others, cardiopulmonary diseases, acute pancreatitis and decompensated liver cirrhosis (Table 2). Among the critically ill patients, there was no association between CTRP1 plasma concentrations and these different disease aetiologies leading to ICU admission (all *p* > 0.05).

### 3.3. CTRP1 Plasma Levels in Critically Ill Patients Are Not Associated with Disease Severity or Mortality

Circulating CTRP1 has been previously suggested as a biomarker for disease severity in various clinical settings [24,25]. Plasma CTRP1 concentrations were not associated with disease severity, as determined by comparing ICU patients with a low (≤10) or high (>10) APACHE-II score. Patients with a high APACHE-II score showed an association between disease severity and elevated CTRP1, but weak evidence at ICU admission (median 768.3 ng/mL vs. median 663.5 ng/mL in APACHE-II ≤10, *p* = 0.339) (Figure 2a). In critically ill patients, who subsequently died during the ICU treatment (*n* = 49), we did not find significantly altered CTRP1 levels at admission to the ICU, indicating that CTRP1 at ICU admission is not a prognostic biomarker in critical diseases. Nevertheless, we observed a tendency towards increased CTRP1 levels in the deceased patients compared to the surviving patients (median 812.4 ng/mL vs. median 711.4 ng/mL in ICU survivors, *p* = 0.166) (Figure 2b).

### 3.4. Elevated CTRP1 Plasma Levels in Critically Ill Patients Are Closely Associated with Diabetic Comorbidity but Not Pre-Existing Obesity

Elevated CTRP1 was previously reported in patients with diabetes. CTRP1 has been associated with alterations in systemic energy metabolism in various conditions [1,12,26]. We therefore assessed whether metabolic comorbidities, including pre-existing obesity or diabetes, impacted CTRP1 levels at ICU admission. Patients with pre-existing diabetes had the highest CTRP1 levels among all ICU patients (*n* = 64, median 833.5 ng/mL vs. median 626.3 ng/mL in non-diabetic patients, *n* = 150, *p* = 0.004; Figure 3a). Moreover, we observed an association between plasma CTRP1 and chronic hyperglycaemia. CTRP1 in critically ill patients displayed a positive correlation with glycated haemoglobin (HbA1c) at ICU admission (r = 0.301, *p* = 0.011) (Table 3), but did not correlate with insulin, glucose or measures of homeostasis model assessment-insulin resistance (HOMA-IR), its reciprocal insulin sensitivity and β-cell function (HOMA-β). In addition, ICU patients with pre-existing obesity, defined as a body mass index (BMI) above 30 kg/m^2^, showed positive correlation between CTRP1 and BMI at ICU admission (r = 0.189; *p* = 0.007) (Table 3, Figure 3c). Despite elevated CTRP1 levels in patients with BMI >30 kg/m^2^, this correlation between BMI and CTRP1 is of weak evidence at ICU admission (*n* = 35, median 798.7 ng/mL, range 286.2–1600.0 ng/mL vs. *n* = 99, median 663.5 ng/mL, range 227.2–1600.0 ng/mL (BMI ≤ 30 kg/m^2^), *p* = 0.219; Figure 3b). Interestingly, we could not find any correlations of CTRP1 to circulating levels of leptin, leptin receptor, adiponectin, ghrelin, resistin and retinol-binding protein 4 (RBP4).

### 3.5. CTRP1 Levels in Critically Ill Patients Are Correlated with Biomarkers of Inflammation, Cholestasis and Renal Failure

We investigated the potential association between CTRP1 and inflammatory response. In agreement with the association between CTRP1 and sepsis, we observed a correlation between CTRP1 and classical markers of inflammation in our cohort, such as interleukin 6 (r = 0.317, *p* < 0.001), procalcitonin (r = 0.414, *p* < 0.001), C-reactive protein (r = 0.238, *p* < 0.001) and suPAR serum levels (r = 0.279, *p* = 0.001; Table 3), an experimental marker of systemic inflammation [27]. We found no correlation to the anti-inflammatory cytokine interleukin 10.

Moreover, we found correlations to markers of renal function including creatinine (r = 0.283, *p* < 0.001, Figure 4a), GFR-cystatin C (r = −0.291, *p* = 0.001) (Figure 4b), urea (r = 0.324, *p* < 0.001) and cystatin C (r = 0.287, *p* = 0.001, Table 3), as well as to markers indicative of cholestasis such as bilirubin (r = 0.422, *p* < 0.033), γ-glutamyltransferase (r = 0.243, *p* < 0.001) and alkaline phosphatase (r = 0.211, *p* < 0.003, Table 3).

In a multivariate logistic regression analysis with CTRP1 and these parameters to test its association with sepsis, only CRP remained an independent and highly significant predictor of sepsis. When we included only creatinine and CTRP1 in the regression model, CTRP1 (and not creatinine) remained independently associated with sepsis (*p* = 0.039), indicating that CTRP1 is associated with sepsis, independent from renal function.

## 4. Discussion

In our study, we demonstrate that CTRP1 plasma levels are significantly upregulated in critical illness as compared to healthy volunteers, displaying highest levels in patients admitted due to sepsis. Despite increased levels of CTRP1 in pre-existing type 2 diabetes and positive correlation with HbA1c in critically ill patients, we did not find an association with insulin resistance or glucose concentration in the blood. CTRP1 levels in our ICU cohort did not correlate with disease severity or short-term ICU survival, but tend to higher CTRP1 levels. In critically ill patients, CTRP1 levels appeared particularly linked to systemic inflammation, metabolic disturbances and organ dysfunction. The potential pathogenic role of CTRP1 in critical illness and sepsis, however, requires further studies. In general, CTRP1 has been associated with changes in systemic energy metabolism in different metabolic and dietary conditions [28,29]. In our cohort of critically ill patients, CTRP1 was indeed related to pre-existing diabetes as well as to long-term blood glucose control reflected by HbA1c, similar to findings in non-ICU patients with diabetes and obesity [14]. Despite the positive correlation of CTRP1 with BMI, but weak evidence, we found that CTRP1 in critical illness is not associated with a variety of established biomarkers of energy substrate metabolism or diabetes-related cytokines such as insulin, leptin, leptin receptor, ghrelin, adiponectin, resistin or retinol-binding protein 4 (RBP4). The explanation of these findings could potentially be assumed in experimental outcomes of animal models of normal and insulin resistant ob/ob mice. On the one hand, elevated CTRP1 levels lowers blood glucose levels without altering insulin or adiponectin levels [2,3], and on the other hand, high CTRP1 concentrations protect from diet-induced obesity and insulin-resistance, while CTRP1 knockout mice developed insulin resistance and hepatic steatosis [6,11]. A further key reason besides its protective role against insulin resistance may be the overlapping inflammatory activity in critically ill patients, which we discuss in the following paragraph. Impaired glucose metabolism in CTRP1 knock out-mice was associated with increased hepatic gluconeogenic gene expression, decreased muscle glucose transporter glucose transporter (GLUT) 4 levels and AMP-activated protein kinase activation [11,12]. Consistent with its effect in mice, CTRP1 was described to act directly and independently of insulin, and to regulate gluconeogenesis in cultured hepatocytes [2]. These expected associations between CTRP1 and metabolic parameters likely reflect the physiological, homeostatic functions of CTRP1 before the onset of critical illness. Our study identified additional, unexpected correlations between elevated levels of CTRP1 and inflammation as well as organ failure in ICU patients. CTRP1 is inversely correlated with the glomerular filtration rate (GFR). In addition, we found further positive correlations with urea, creatinine and cystatin C. This association of CTRP1 to renal impairment might be a possible reason for elevated CTRP1 in circulation. This could be due to the retention of CTRP1 in patients with renal insufficiency. CTRP1 correlated significantly with indicative markers of liver damage and cholestasis. These results are presumably related to the corresponding correlation results with parameters of glucose and energy metabolism as well as inflammatory response.

Experimental studies indeed support that CTRP1 is involved in controlling systemic inflammatory responses. In lipopolysaccharide (LPS) stimulated rats, CTRP1 is expressed at high levels in adipose tissues [8]. The LPS-induced increase in CTRP1 gene expression was found to be mediated by inflammatory cytokines including tumor necrosis factor-α and interleukin 1β [8,30]. CTRP1 itself causes a concentration-dependent expression of further inflammatory markers in endothelial and vascular smooth muscle cells [1,19,31], thereby serving as an amplifier of inflammation beyond its tissue of origin. For instance, in heart failure CTRP1 levels in epicardial adipose tissue and blood circulation are elevated [8]. Elevated CTRP1 levels in circulation increased IL-6 mRNA levels in congestive heart failure and induced aldosterone release through JAK2-STAT3 signaling pathways [8,30,32]. Similar mechanisms may be present in patients with sepsis, where disturbed blood pressure, vascular tone and vascular barrier dysfunction are commonly observed [31,33]. On the other hand, CTRP1 administration in experimental CTRP1 knockout mice protected against hyper-inflammation through activation of the S1P/cAMP signalling pathways in cardiomyocytes [6,32]. We found a strong correlation of CTRP1 with classical inflammatory parameters such as CRP, procalcitonin and IL-6. Along with the positive correlation between CTRP1 and suPAR, CTRP1 might be induced by the activated immune system [27]. Despite the close correlation between CTRP1 and inflammatory markers in our study, elevated CTRP1 did not predict an adverse prognosis at the ICU. Thus, further studies are warranted to clarify whether elevated CTRP1 reflect an either harmful or protective endogenous response to critical illness.

In summary, CTRP1 is correlated with several factors such as chronic hyperglycaemia (HbA1c), inflammation (CRP), renal function (creatinine) and liver injury (bilirubin), making it challenging to dissect its clinical relevance from confounding factors. However, although our study included 218 patients, the sample size of our study very likely does not allow to control for all potential confounders.

## 5. Conclusions

CTRP1 is a promising novel molecular mediator connecting inflammatory and metabolic diseases [3,4]. In our large, prospectively enrolled study, CTRP1 levels were significantly elevated in critically ill patients and were associated with inflammation and sepsis as well as diabetes and metabolic disturbances. Our data support that CTRP1 is integrated in the complex network of adipokines in the pathogenesis of critical illness, sepsis and organ failure. However, CTRP1 does not yet have a clear clinical benefit, but the recognition of potential correlation between metabolic changes and inflammation in intensive care patients could indicate a clinical relevance of this adipokine. Mechanistic studies are needed to clarify its exact pathogenic role in this setting. A more detailed knowledge of the different roles of CTRP1 in metabolic and inflammatory pathways may result in a better understanding of critical disease conditions.

## Figures and Tables

**Figure 1 jcm-08-00661-f001:**
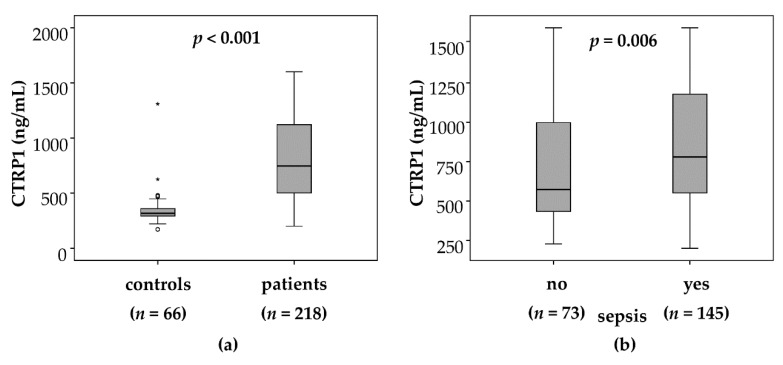
CTRP1 levels in critically ill patients and sepsis. (**a**) CTRP1 plasma concentrations, at time of admission to the ICU, were significantly elevated in critically ill patients (*n* = 218) compared with healthy controls (*n* = 66) (*p* < 0.001; U-Test). (**b**) CTRP1 levels are associated with the presence of sepsis (sepsis, *n* = 145; no sepsis, *n* = 73) (*p* = 0.006; U-Test). *: extreme outlier; ICU—intensive care unit; CTRP1—C1q/TNF-related protein 1.

**Figure 2 jcm-08-00661-f002:**
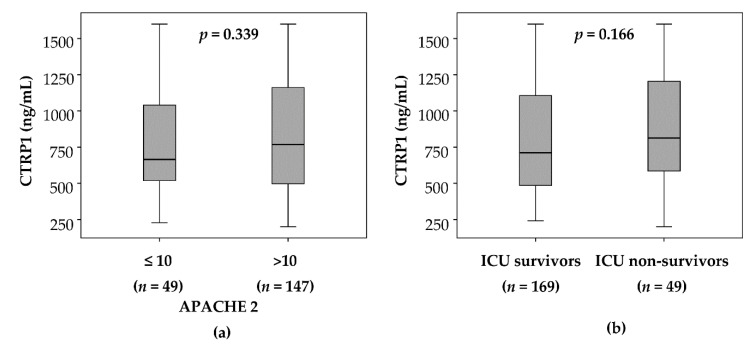
CTRP1 levels in critically ill patients are not associated with disease severity and short-term mortality. (**a**) Patients with high disease severity (*n* = 147), as defined by an APACHE-II score above 10, are not associated with elevated plasma CTRP1, but show a tendency towards higher CTRP1 levels at ICU admission (*p* = 0.339; U-Test). (**b**) Patients that died during the course of ICU treatment (*n* = 49) are characterized by a tendency towards higher plasma CTRP1 concentrations already at ICU admission (*p* = 0.166; U-Test). APACHE—Acute Physiology And Chronic Health Evaluation; ICU—intensive care unit; CTRP1—C1q/TNF-related protein 1.

**Figure 3 jcm-08-00661-f003:**
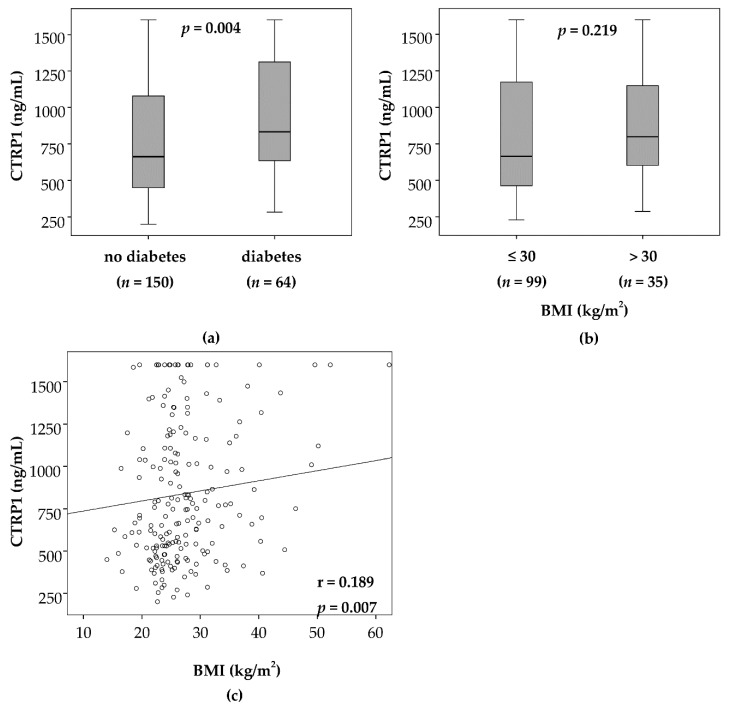
Impact of metabolic comorbidities on CTRP1 levels. CTRP1 plasma concentrations are significantly elevated in ICU patients with pre-existing type 2 diabetes (*n* = 150) (*p* = 0.004; U-test) (**a**). CTRP1 levels are not associated with obesity, as defined by a body-mass index (BMI) above 30 kg/m^2^ (*n* = 55) (*p* = 0.219; U-Test (**b**) and r = 0.189, *p* = 0.007; Spearman rank correlation test (**c**). BMI—body mass index; ICU—intensive care unit; CTRP1—C1q/TNF-related protein 1.

**Figure 4 jcm-08-00661-f004:**
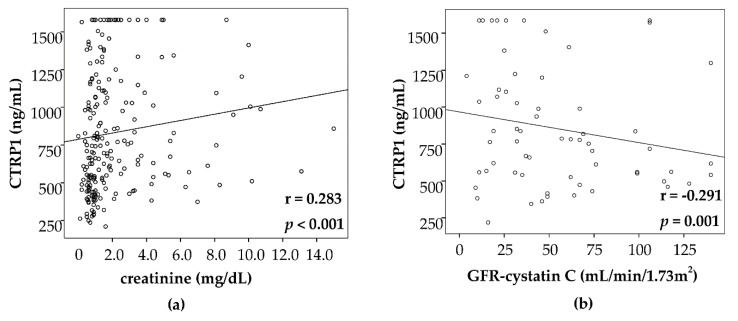
CTRP1 levels correlate with renal function in critically ill patients. In the ICU cohort, CTRP1 correlates with markers of excretory renal function such as creatinine (**a**) and GFR-cystatin C (**b**). GFR—glomerular filtration rate.

**Table 1 jcm-08-00661-t001:** Baseline patient characteristics and CTRP1 plasma measurements.

Parameter	All Patients	Non-Sepsis	Sepsis	* *p*
Number n	218	73	145	
Sex (male/female) n	133/85	48/25	85/60	n.s.
Age (years)	64 (18–90)	61 (18–85)	65 (20–90)	n.s.
APACHE-II score	18 (2–43)	13.5 (2–33)	19 (4–43)	<0.001
ICU days	7 (1–137)	6 (1–45)	9 (1–137)	0.004
Death during ICU n (%)	49 (22.5%)	9 (12.3%)	40 (27.6%)	0.010
Death during follow-up (total) n (%)	89 (40.8%)	22 (30.1%)	67 (46.2%)	0.026
Mechanical ventilation n (%)	143 (65.6%)	46 (63%)	97 (66.9%)	n.s.
Pre-existing diabetes n (%)	64 (29.4%)	22 (30.1%)	42 (29.0%)	n.s.
BMI (m^2^/kg)	25.8 (14–86)	25.7 (15.9–40.5)	28.9 (14–86.5)	n.s.
WBC (×10^3^/µL)	13.1 (0.1–208)	12.5 (1.8–29.6)	14.1 (0.1–208)	0.024
CRP (mg/dL)	100.5 (5–230)	17 (5–230)	164 (5–230)	<0.001
IL-6 (pg/mL)	150.0 (2–28000)	66.5 (1.5–5000)	250 (0.1–28000)	<0.001
Procalcitonin (ng/mL)	0.7 (0.03–207.5)	0.2 (0.03–100)	2.2 (0.1–207.5)	<0.001
Creatinine (mg/dL)	1.3 (0.1–15)	1.0 (0.2–15)	1.5 (0.1–10.7)	0.017
GFR-Cystatin C (mL/min)	34 (0–379)	59 (5–379)	21.5 (0–218)	<0.001
INR	1.16 (0.92–13)	1.17 (0.95–6.73)	1.16 (0.92–13)	n.s.
CTRP1 day 1 (ng/mL)	747.1 (200.5–1600)	574.2 (227.2–1600)	779.6 (200.5–1600)	0.006

For quantitative variables, median and range (in parenthesis) are given. Percentages in parenthesis refer to the total number of patients in the respective groups. * Significance between sepsis and non-sepsis patients was assessed using the Mann–Whitney-U-test (for quantitative variables) or the chi-square test (for categorical variables); n.s.—not significant. APACHE—Acute Physiology And Chronic Health Evaluation; BMI—body mass index; CRP—C-reactive protein; IL-6—interleukin 6; ICU—intensive care unit; INR—international normalized ration; WBC—white blood cell.

**Table 2 jcm-08-00661-t002:** Disease aetiology of the study population leading to ICU admission.

	Sepsis	Non-Sepsis
	*n* = 145	*n* = 73
**Aetiology of sepsis critical illness**		
Site of infection n (%)		
Pulmonary	72 (50%)	
Abdominal	28 (19%)	
Urogenital	11 (8%)	
Other	34 (23%)	
**Aetiology of non-sepsis critical illness** n (%)		
Cardio-pulmonary disorder		29 (40%)
Acute pancreatitis		10 (14%)
Acute liver failure		4 (5.5%)
Decompensated liver cirrhosis		9 (12%)
Severe gastrointestinal hemorrhage		4 (5.5%)
Non-sepsis other		17 (23%)

**Table 3 jcm-08-00661-t003:** Correlations with CTRP1 plasma concentrations at ICU admission day.

	ICU Patients	
Parameters	r	*p*
Obesity/diabetes		
BMI	0.189	0.007
HbA1c	0.301	0.011
Inflammatory response		
CRP	0.238	<0.001
IL-6	0.317	<0.001
PCT	0.414	<0.001
suPAR	0.279	0.001
Renal function		
Urea	0.324	<0.001
Creatinine	0.283	<0.001
Cystatin C	0.287	0.001
GFR Cystatin C	−0.291	0.001
Liver injury/cholestasis		
Bilirubin	0.422	<0.001
GLDH	0.154	0.033
γ-GT	0.243	<0.001
AP	0.211	0.003

Spearman rank correlation test, only statistically significant results are shown. The overall weak associations may not be clinically relevant. However, the purpose of the statistical correlation analysis is to descriptively discuss the clinical relevance of CTRP1. Understanding these aspects will help better utilize the evidence to improve clinical decision-making. BMI—body mass index; HbA1c—hemoglobin A1c; CRP—C-reactive protein; IL-6—interleukin 6; PCT—procalcitonin; suPAR—soluble urokinase-type plasminogen activator receptor; GFR—glomerular filtration rate; GLDH—glutamate dehydrogenase; γ-GT—gamma-glutamyltransferase; AP—alkaline phosphatase.

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
