# Peer review of "Elevated CTRP1 Plasma Concentration Is Associated with Sepsis and Pre-Existing Type 2 Diabetes Mellitus in Critically Ill Patients"

_jcm, 2019, doi:10.3390/jcm8050661_

Reviewer 1 Report

In the introduction (lines 70-73) authors state that the study was performed to learn about the value of CTRP1 “as a biomarker” and to investigate its “diagnostic and clinical relevance”. The discussion lacks any clear statement about usefulness as a biomarker or diagnostic/clinical relevance. If there is no evidence for clinical usefulness (which I would say is true), this should be clearly stated in the discussion. If you see evidence for usefulness in the clinic, please state and argue this.

The healthy control group needs better description. There is only a reference given in line 82, presumably to indicate that data from this group has already been used in another publication. It appears that the reference is wrong and should be [18], not [19], and unfortunately I do not have access to this preceding publication. But please consider:

(1.) Characteristics of the healthy controls should, as far as available, also be shown in Table 1 – it needs to be comprehensible, for which parameters healthy controls were matched with the patient groups.

(2.) Should some of the critically ill patients described in the submitted manuscript also be identical with patients from the previous publication, this must be clearly stated in order to avoid any impression of violating a rule of double publication.

According to Table 1 you had data from 218 patients in total, but the total number in several figures is lower (Fig.2a, Fig.2a,b). Please explain.

I do not understand the %-values for “sepsis” and “non-sepsis” in Table 1. E.g. for diabetes: If you take the 64 diabetic patients as 100%, then the % values for “sepsis” and “non-sepsis” must sum up to 100%. If you take all 218 patients as 100%, then the % values for “sepsis” and “non-sepsis” must sum up to 29.4%? Neither is the case. Also applies to other parameters. Please explain.

It seems that CTRP1 concentrations >1600 ng/ml could not be measured and that all values >1600 entered statistics as “1600”. If so, this needs to be stated in the Experimental Section. Re-measurement of diluted samples could strengthen the results.

Addressing observed differences as “trends” is comprehensible with p values up to 0.1 or perhaps 0.15. But any interpretation (except in the sense of no evidence for a difference between the groups) appears overshooting  with p-values >0.3.

Findings from associations appear to require cautious interpretation. While p-values<0.05 indicate that these parameters are in principle associated, accompanying r values <0.3 or even <0.2 implicate that this association is very very weak (as also obvious from the dot blots). Please seriously consider that very weak associations are unlikely to reflect an important pathophysiological mechanism or usefulness as a biomarker. This needs to be openly discussed.

Lines 93,94: Please explain exactly what this means – what are “outside values” and “far out values”?

Lines 121 and 128: Please check, whether the “<” is="" correct.="" it="" appears="" that="" should="" be="">” in both cases.  

Why not add statistical information (p-values) in Table 1?

Line 148: It is not clear, what “glycemic controls” refers to.

Fig.1a: What do the full and empty dots in the control graph mean?

Line 154: Give range for both groups.

In the discussion only a very minor number of sentences relates to results and findings from the present study. Most of the discussion just summarizes findings from previous publications.

Author Response

Response to Reviewer 1 Comments

Point-by-point response to the reviewer:

Thank you very much for the thorough and fair review of our manuscript.

Reviewer #1:

In the introduction (lines 70-73) authors state that the study was performed to learn about the value of CTRP1 “as a biomarker” and to investigate its “diagnostic and clinical relevance”. The discussion lacks any clear statement about usefulness as a biomarker or diagnostic/clinical relevance. If there is no evidence for clinical usefulness (which I would say is true), this should be clearly stated in the discussion. If you see evidence for usefulness in the clinic, please state and argue this.

Response:

We thank the reviewer for this excellent remark. We fully agree with the referee that the term “biomarker” at this point positions CTRP1 hastily as a diagnostic tool and we have deleted it from the introduction section (see page 2).

We therefore would like to emphasize the exploratory nature of our work. Our intention is to describe the potential interdependence of this adipocytokine in intensive care patients with regard to metabolism and inflammation and to discuss the diagnostic and clinical relevance of CTRP1 as stated in the conclusion. Our findings contribute to the understanding of CTRP1 as a useful tool for improved and deeper understanding the complex network of metabolic and inflammatory changes in intensive care patients and their associated clinical consequences, e.g. survival and severity of disease. However, further studies on this issue are needed to clarify its exact pathogenic role in this setting.  

CTRP1 does not have a predictive characteristic in our cohort with respect to the prognosis of intensive care patients. Therefore, in the conclusion, we have clearly emphasized once again that CTRP1 does not yet have a clear clinical benefit, but the recognition of the potential correlation between metabolic changes and inflammation in intensive care patients with CTRP1 potentially indicate a clinical relevance of this adipocytokine (see pages 9 and 11, lines 287-290).

The healthy control group needs better description. There is only a reference given in line 82, presumably to indicate that data from this group has already been used in another publication. It appears that the reference is wrong and should be [18], not [19], and unfortunately I do not have access to this preceding publication.

Response:

We thank the reviewer for his very careful attention. We sincerely apologize for this error and corrected this typo. In fact, in lines 86 to 88, instead of listing references 21 to 23, the previous version remained these references at No. 17 to 19. We have now very thoroughly and accurately adjusted the numbering of the references throughout the manuscript and assure the correct presentation of the references.

But please consider:

(1.) Characteristics of the healthy controls should, as far as available, also be shown in Table 1 – it needs to be comprehensible, for which parameters healthy controls were matched with the patient groups.

Response:

We appreciate this comment and have added the requested information about the healthy control group (age, sex, BMI) in the revised manuscript (experimental section, see page 2, lines 88-90).

(2.) Should some of the critically ill patients described in the submitted manuscript also be identical with patients from the previous publication, this must be clearly stated in order to avoid any impression of violating a rule of double publication.

Response:

We appreciate this comment and have provided additional information in the experimental section. The study design has been published before. The current cohort of patients was collected from an ongoing, prospective observational trial in our ICU at the RWTH University hospital, in which patients were included consecutively. For the current analysis, we randomly enrolled n=218 patients that had been treated between 2006 and 2011 from the existing biobank. We have given this information in the experimental section (see page 2, lines 81-84).

According to Table 1 you had data from 218 patients in total, but the total number in several figures is lower (Fig.2a, Fig.2a,b). Please explain.

Response:

We thank the referee for this comment. In Figures 2a and 3a,b the sum of 218 patients is not reached. Due to the data available from the biodatabase, no corresponding Information was available for a few patients, so that these could not be taken into account for certain analyses. Data on APACHE, diabetes and BMI are therefore not available for all patients.

I do not understand the %-values for “sepsis” and “non-sepsis” in Table 1. E.g. for diabetes: If you take the 64 diabetic patients as 100%, then the % values for “sepsis” and “non-sepsis” must sum up to 100%. If you take all 218 patients as 100%, then the % values for “sepsis” and “non-sepsis” must sum up to 29.4%? Neither is the case. Also applies to other parameters. Please explain.

Response:

We appreciate this comment by the referee and have provided additional information in the table 1 of the revised manuscript. However, the percentages refer to the total number of “non-sepsis” (N=73) and “sepsis” (N=145) patient groups, but also to “all patients” (N=218) indicated in row 1 of Table 1 (see page 4, lines 124-125).

It seems that CTRP1 concentrations >1600 ng/ml could not be measured and that all values >1600 entered statistics as “1600”. If so, this needs to be stated in the Experimental Section. Re-measurement of diluted samples could strengthen the results.

Response:

We sincerely thank the referee for this suggestion and have provided additional information in the experimental section. Unfortunately, a limited volume of plasma samples of the patients was available and the dilution factor was used in all samples. Despite this, a higher dilution could not be performed in a few patients with results higher than the higher linearity limit of the ELISA (see page 3, lines 98-101).

Addressing observed differences as “trends” is comprehensible with p values up to 0.1 or perhaps 0.15. But any interpretation (except in the sense of no evidence for a difference between the groups) appears overshooting  with p-values >0.3.

Response:

We agree with the referee that the term “trend” is imprecise and have deleted it from the results section. We provided the addition of “…an association between X and Y but weak evidence…”. Nevertheless, trends in absolute values should be mentioned in order to be able to draw cautious and preliminary assumptions if necessary (see pages 5-7, results section subheadings 3.3, 3.4 and figure 2-, table- 3 legends)

Findings from associations appear to require cautious interpretation. While p-values<0.05 indicate that these parameters are in principle associated, accompanying r values <0.3 or even <0.2 implicate that this association is very very weak (as also obvious from the dot blots). Please seriously consider that very weak associations are unlikely to reflect an important pathophysiological mechanism or usefulness as a biomarker. This needs to be openly discussed.

Response:

We thank the reviewer for raising this important aspect. We agree that the diagnostic and clinical relevance might be premature at this stage of this explorative statistical analysis. We therefore would like to emphasize the exploratory nature of our work. We have stressed in the table 3 legend in our revised manuscript that the overall weak associations may not be clinically relevant. However, the purpose of these statistical correlation analysis is to descriptively discuss the unknown clinical relevance of CTRP1. Understanding these aspects will help better utilize the evidence to improve clinical decisionmaking (see table- 3 legend, page 7).

Lines 93,94: Please explain exactly what this means – what are “outside values” and “far out values”?

Response:

We appreciate this comment and have provided additional information in the experimental section of the revised manuscript (see page 3, lines 103-109).

Lines 121 and 128: Please check, whether the “<” is="" correct.="" it="" appears="" that="" should="" be="">” in both cases.  

Response:

We thank the reviewer for his very careful attention. We sincerely apologize for this error and corrected this typo in the first of these two mentioned aspects (see page 4, line 144). 

Why not add statistical information (p-values) in Table 1?

Response:

We appreciate this comment by the referee. We add statistical information about the p-values in table 1 (see pages 3 and 4, table 1 and its legend).

Line 148: It is not clear, what “glycemic controls” refers to.

Response:

We are sorry for the misunderstanding, which is likely due to the fact that CTRP1 in critically ill patients displayed a statistically positive correlation with glycated haemoglobin (HbA1c) at ICU admission. We replaced the term “glycemic controls” by the term “chronic hyperglycaemia” (see page 6, line 172).

Fig.1a: What do the full and empty dots in the control graph mean?

Response:

We thank the referee for this comment. The empty dots in figure 1a represent the outliers and the full dots are representing the extreme (far out) values.

Line 154: Give range for both groups.

Response:

We sincerely thank the referee for this suggestion. Ranges for both groups have been added for CTRP1 plasma levels in patients with BMI lower or higher than 30 kg/m² (see page 6).

In the discussion only a very minor number of sentences relates to results and findings from the present study. Most of the discussion just summarizes findings from previous publications.

Response:

We appreciate this comment and have provided additional information in the discussion section (see page 8, lines 224-227 and page 9, lines 251-257 and 271-273).

Reviewer 2 Report

The paper by Yagmur et al has investigated the circulating levels of CTRP in sepsis and critical illness. The main question of the paper is clear and is addressed by the experiments performed. 

This paper has carried out a scientifically sound study, although the originality of the findings are limited. the Approach is appropriate.

However this reviewer would like to clarify some points in the publication:

-The reasoning behind investigating type-2 diabetes is not clearly addressed in the abstract nor throughout the manuscript. The justification of looking at this condition in particular should be addressed in the introduction.

-Figure legends are not descriptive, they are lacking crucial information, notably the sample size, details of statistical tests used and explaining abbreviations. Figure legends may be improved for clarity (details of tests used, etc).

-Some data are incoherent with previously published results from other groups. Notably well characterised variables such as adiponectin and leptin that correlate to BMI do not correlate with CTRP in this study. This is clearly stated in the results, but should be addressed in the discussion.

-In the methods, authors describe that all outliers were included. Data controlling for outliers should be investigated and should be at least placed in supplementary materials.

-This study is largely ‘epidemiological’ in its approach, this reviewer proposes that correlations and associations should be recalculated whilst controlling for confounding factors. E.g. controlling for CRP, cytokine levels, adiponectin levels, BMI and HbA1c (now that we know the latter two factors correlate to CTRP levels)

-Do results still stand when controlling for the above?

Author Response

Point-by-point response to the reviewer:

Thank you very much for the thorough and fair review of our manuscript.

Reviewer #2:

The paper by Yagmur et al has investigated the circulating levels of CTRP in sepsis and critical illness. The main question of the paper is clear and is addressed by the experiments performed. 

This paper has carried out a scientifically sound study, although the originality of the findings are limited. the Approach is appropriate.

However this reviewer would like to clarify some points in the publication:

-The reasoning behind investigating type-2 diabetes is not clearly addressed in the abstract nor throughout the manuscript. The justification of looking at this condition in particular should be addressed in the introduction.

Response:

We sincerely thank this expert reviewer for his/her positive evaluation and the encouraging comments. We addressed in the abstract and in the introduction the reasoning why investigating CTRP1 in critically ill patients as a potential link between metabolic and inflammatory alterations (see page 1, lines 21-23 and page 2, lines 72-75).

-Figure legends are not descriptive, they are lacking crucial information, notably the sample size, details of statistical tests used and explaining abbreviations. Figure legends may be improved for clarity (details of tests used, etc).

Response:

We appreciate this comment and have provided additional information in the figure legends, notably sample size, details of statistical tests used and explaining abbreviations.

-Some data are incoherent with previously published results from other groups. Notably well characterised variables such as adiponectin and leptin that correlate to BMI do not correlate with CTRP in this study. This is clearly stated in the results, but should be addressed in the discussion.

Response:

We thank the reviewer for stressing out this important aspect. In fact, to address this issue more precisely, we have concretized our discussion. The explanation of this aspect raised by the referee could potentially be assumed in experimental outcomes of animal models of normal and insulin resistant ob/ob mice. On the one hand elevated CTRP1 levels lowers blood glucose levels without altering insulin or adiponectin levels and on the other hand high CTRP1 concentrations protect from diet-induced obesity and insulin-resistance, while CTRP1 knockout mice developed insulin resistance and hepatic steatosis. A further key reason besides its protective role against insulin resistance may be the overlapping inflammatory activity in critically ill patients which we also discussed (see pages 8-9, lines 231-244).

-In the methods, authors describe that all outliers were included. Data controlling for outliers should be investigated and should be at least placed in supplementary materials.

Response:

We appreciate this comment by the referee. However, as we mentioned in the experimental section, due to the skewed distribution of the parameters, data are given as median and range, and shown graphically by box-and-whiskers plots. Given that the outliers are data points lying far away from the majority of other data points, outliers in the data that is not normally distributed do not require identification. The median and range, we used statistics that are less sensitive to outliers. In addition, box plots such as in figure 1a are used to identify the outliers. In this box plot, any data that lies outside the upper or lower fence lines is considered outliers.

-This study is largely ‘epidemiological’ in its approach, this reviewer proposes that correlations and associations should be recalculated whilst controlling for confounding factors. E.g. controlling for CRP, cytokine levels, adiponectin levels, BMI and HbA1c (now that we know the latter two factors correlate to CTRP levels)

-Do results still stand when controlling for the above?

Response:

We fully agree with the reviewer that CTRP1 is correlated with several factors, making it challenging to dissect its clinical meaning from confounding factors. For instance, CTRP1 is correlated with obesity (BMI), inflammation (CRP), renal function (creatinine) and liver injury (bilirubin), as depicted in table 3. When we performed a multivariate logistic regression analysis with CTRP1 and these parameters to test its association with sepsis, only CRP remained an independent and highly significant predictor of sepsis. When we included only creatinine and CTRP1 in the regression model, CTRP1 (and not creatinine!) remained independently associated with sepsis (p=0.039), indicating that CTRP1 is associated with sepsis, independent from renal function. However, although our study included 218 patients, the sample size of our study very likely does not allow to control for all potential confounders. We therefore would like to emphasize the exploratory nature of our work and have stressed this in the revised manuscript (results section, subheading 3.5, page 8, lines 212-216) and discussion section, page 9, lines 277-281).

Round  2

Reviewer 1 Report

I appreciate your serious efforts and I believe that the paper has considerably improved.

There is one remaining point that I regard as important: You now appropriately explain in lines 98 to 101 that you had several values above the linearity range. But it is necessary that you also state clearly, how you proceeded with these values, i.e. that all of them entered statistics as “1600”. (Taking all these values as “1600” unavoidably afflicts your statistics with a certain bias, but if you cannot re-measure, any other procedure would likewise be problematic – the point is that the reader needs to be informed, why all whiskers end exactly at 1600 and why the dots accumulate at 1600 in the dot plots.)

Author Response

Response to Reviewer 1 Comments

Round 2

Response to the reviewer:

Thank you very much for the thorough and fair review of our manuscript.

Reviewer #1:

I appreciate your serious efforts and I believe that the paper has considerably improved.

Response:

We thank the reviewer for her/his thorough and positive evaluation of our work.

There is one remaining point that I regard as important: You now appropriately explain in lines 98 to 101 that you had several values above the linearity range. But it is necessary that you also state clearly, how you proceeded with these values, i.e. that all of them entered statistics as “1600”. (Taking all these values as “1600” unavoidably afflicts your statistics with a certain bias, but if you cannot re-measure, any other procedure would likewise be problematic – the point is that the reader needs to be informed, why all whiskers end exactly at 1600 and why the dots accumulate at 1600 in the dot plots.)

Response:

We would like to thank the reviewer for her/his thoughtful comment. The reviewer’s assumption is indeed correct. All samples that have been measured with values above the linearity range were set to 1600 ng/mL. While this method ensures that all samples were included in all statistical tests, we acknowledge that the lack of exact values in the upper 10% of the samples might underestimate the performance of the biomarker. Nonetheless, to our opinion, this procedure minimizes the risk of accidental overinterpretation of the data. These details are now included in the revised manuscript (see page 3, lines 101-104).
